# Developmental Dysplasia of Hip: Perspectives in Genetic Screening

**DOI:** 10.3390/medsci7040059

**Published:** 2019-04-11

**Authors:** Radoslav Zamborsky, Milan Kokavec, Stefan Harsanyi, Doaa Attia, Lubos Danisovic

**Affiliations:** 1Department of Orthopaedics, Faculty of Medicine, Comenius University and National Institute of Children’s Diseases, Limbova 1, 833 40 Bratislava, Slovakia; milan.kokavec@hotmail.com; 2Institute of Medical Biology, Genetics and Clinical Genetics, Faculty of Medicine, Comenius University, Sasinkova 4, 811 08 Bratislava, Slovakia; stefan.harsanyi@fmed.uniba.sk (S.H.); lubos.danisovic@fmed.uniba.sk (L.D.); 3Department of Internal Medicine, Faculty of Medicine, Alexandria University, Chamblion St., Azarita, 21131 Alexandria, Egypt; doaa.attia@gmail.com

**Keywords:** developmental dysplasia of hip, DDH screening, genetic factors, associated genes

## Abstract

Development dysplasia of the hip (DDH) is a complex developmental disorder despite being a relatively common condition mainly caused by incompatibility of the femoral head and the abnormal joint socket. Development dysplasia of the hip describes a wide spectrum of disorders ranging from minor acetabular dysplasia to irreducible dislocation of the hip. Modern medicine still suffers from lack of information about screening and precise genetic examination. Genome wide linkage and association studies have brought significant progress to DDH diagnosis. Association studies managed to identify many candidate (susceptible) genes, such as *PAPPA2*, *COL2A1*, *HOXD9*, *GDF-5*, and *TGFB1*, which play a considerable role in the pathogenesis of DDH. Early detection of DDH has a big chance to help in preventing further disability and improve the psychological health and quality of life in those children. This emphasizes the importance to establish a universal screening program along with the genetic counseling.

## 1. Introduction

Developmental dysplasia of the hip (DDH) is a complex musculoskeletal disease, which represents a wide spectrum of pathology, ranging from an asymptomatic form, with only mild radiological deviations, to minor joint instability, acetabular dysplasia, subluxation and to irreducible hip dislocation [1,2]. Development dysplasia of the hip could be an isolated disease or associated with other conditions such as club feet, cardiac anomalies, and renal problems. Untreated DDH can lead to secondary damage to femur, destruction of the joint cartilage and, later on, even to severe movement impairment at any age [3,4].

Persistent DDH might lead to premature osteoarthritis (OA) in adulthood [5]. Kolundžić et al. [6] conducted a prospective cohort study confirming that DDH is a major risk factor to develop severe OA that requires arthroplasty. Similar situations occur in mild forms of DDH that can be missed in newborns, and without treatment, can lead to earlier senior OA, for which the end-stage treatment could be total hip replacement.

Genetic components play crucial roles in the pathogenesis of DDH. Genetic risk exhibited a large familial segregation [1,7]. Studies reported a 30-fold increase for DDH among siblings and children with a previous family history of DDH. This strong genetic basis has been confirmed in a twin-concordance study with a higher incidence among monozygotic (41%) compared to dizygotic twins, and with 5% recurrence rate for the subsequently born children [8]. 

## 2. Diagnosis and Pathology

Historic data shows that hip dysplasia has been known since ancient times (Hippocrates) and with technological and theoretical advancement through the centuries, from which ultrasonography was the most helpful, researchers were able to define this disorder and prepare methods to treat it. Notable names from the 20th century such as Marino Ortolani (Ortolani’s maneuver, pillow method) or Bedrich Frejka (modificated pillow method) are connected with diagnostic and treatment methods that are still in use [9].

Development dysplasia of the hip can be diagnosed in childhood, but also in adults. Statistically, it affects mainly females with a sex ratio of 5:1. Higher occurrence of DDH in females is in some literature said to be caused by relaxin, a hormone which affects laxity of pelvic ligaments in the mother and has stronger effect on the female fetus, but this theory needs to be further explored. Left hip is more affected in 37% of patients compared to 26.5% in bilateral affection [10]. Incidence of DDH varies greatly with countries and continents, but an estimated incidence of 1/34 cases of DDH/1000 live births was reported worldwide. However, the incidence is higher in certain areas such as Middle East, gulf area, and Italy [1,10]. 

This variation in the different geographical locations is due to the difference in the diagnostic modalities and timing of evaluation [1,11]. This is also the reason of controversy in the literature regarding DDH research data such as prevalence, incidence and diagnosis. Moreover, the high consanguinity rate in the Gulf area attributes to such a high incidence of DDH as well [1,4,12].

The etiology of DDH is multifactorial. It consists of genetic, environmental, and mechanical risk factors. The environmental and mechanical causes include high birth weight (HBW), breech presentation, oligohydramnios, primiparity, intrauterine malposition, swaddling, and laxity of ligaments [3,10,13]. 

Studies conducted in Asia reported the existence of genes which could be associated with this disorder. Shi et al. [2] reported the possible correlation between the pregnancy-associated plasma protein A2 (PAPPA-2) and DDH among the Chinese Han population. Recently, exome sequencing and genome linkage studies in large families segregating DDH have been useful to identify some variants in the genes involved in chondrogenesis and bone formation that have been reported to be associated with DDH such as collagen alpha-1(I) chain gene (*COL1A1*) and vitamin D receptor (VDR) [14].

Despite the lack of standardized screening methods to detect DDH in early age, neonatal physical examination and ultrasonography (U/S) play a crucial role to detect early conditions and decrease the risk of complications (Figure 1, Figure 2 and Figure 3). Approximately 97% of orthopedic surgeons use U/S to diagnose DDH before the age of three, however, pelvic radiography is a better imaging tool for children after the age of 3 years [15].

The future in treatment is early screening, diagnosis and treatment. Without reliable genetic examination there cannot be an effective approach to early therapy. The goal of treatment is to obtain a safe reduction of the hip conservatively and to avoid surgical intervention and long-term disabilities. In this review, we provide a better understanding of DDH and bridge the gap between genetic causes and molecular mechanisms.

## 3. Study Designs

### 3.1. Genome Wide Association Studies 

Genome Wide Association Studies (GWAS) are systematic observational studies, which analyze genome-wide set of genetic variants preferably in a large number of different individuals to determine if any of the observed variants are associated with a trait. Genome Wide Association Studies usually focuses on the correlation between single nucleotide polymorphisms (SNPs) to identify variants in the susceptible genes associated with certain diseases [1]. 

Several musculoskeletal disorders were discovered through GWAS. Sun et al. [16] conducted a large GWAS to study DDH and found 12 variants in the ubiquinol-cytochrome C reductase complex chaperone (*UQCC*) gene to be associated with DDH. Ubiquinol-cytochrome C reductase complex chaperone plays a role in spine and bone formation. 

Many genes of interest have been found to be associated with DDH such as *COL1A1, GDF5, HOXB9, ASPN, VDR, IL6-gene,* and *HOXD9* [6,10,17,18,19,20,21,22]. Those genes are found to help the bone formation and chondrogenesis in large case-control studies among the Chinese and Caucasian populations (Table 1). 

Genome Wide Association Studies are used to find genome wide or candidate associations, out of which the Candidate Gene Association Analyses (CGAA) are more important and widely studied in the context of developmental dysplasia of the hip.

### 3.2. Genome Wide Linkage Analysis

Genome Wide Linkage Analyses (GWLA) are systematic studies, that utilize pedigrees to identify loci associated with a disease and are also used in genetic mapping of traits with genetic predisposition. Genome Wide Linkage Analyses are based on the tendency, that two loci are inherited together. These studies demonstrate several chromosome region segregations with trait phenotype in large family pedigree [25].

Reviewing the literature showed that large pedigree of families with DDH had a 4 Mb region on chromosome 17q21.32 [25]. Moreover, the analysis of large multigenerational families of DDH showed a linkage to chromosomes 4q35, 13q22, and 16p [23,27]. A GWAS study from a large four generation Japanese family with history of acetabular dysplasia found a linkage of DDH with specific regions at chromosome 13 [26].

### 3.3. Global Copy Number Variants Detection 

DNA sequence diversity within the human genome is said to be more affected by Global Copy Number Variants (CNVs) than SNPs. The importance of CNVs in genome wide association studies is becoming widely accepted, but the optimal method to identify these variants is still in question. Older methods e.g., fluorescence in situ hybridization (FISH) lacked in genetic resolution and could identify only large repeats. Recently, high throughput genomic sequencing method has great potential to enhance CNV diagnostic. Copy number variants are submicroscopic genomic alterations that might affect a considerable number of DNA base pairs either by duplication or deletion [28]. Copy Number Variants are involved in the disease by changing the copy numbers of dosage sensitive genes or disrupting the gene coding regions [1,29]. Moreover, non-allelic homologous recombination and end-joining were reported in the genomic rearrangement and formation of CNVs [29]. CNVs have been studied in Mendelian diseases and neurological diseases such as Schizophrenia and autism [29,30]. However, it is not implicated yet in DDH. Therefore, such a step would lead to very promising findings to better understanding the molecular and genetic basis of DDH [31].

### 3.4. Whole Exome Sequencing 

Whole Exome Sequencing (WES) is a genomic method for sequencing used to sequence all protein coding portions (exome) of the genome and to identify variants in all individual genes [1]. This method can be applied to both clinical and research diagnostic, because these variants are responsible for polygenic and also Mendelian diseases. WES has been used to identify several hereditary and musculoskeletal abnormalities [24,32]. Such techniques had a novel impact on identification of variant involvement in DDH, e.g., Feldman et al. [31] used WES in large pedigree segregating DDH and identified those variants in chemokine receptor genes (CX3CR1) was implicated in the pathology of DDH. Both linkage analysis studies and exome data supported the association of CX3CR1 and DDH (Table 1). Zhu et al. [33] found, that the ribosome biogenesis factor (BMS1) is associated with DDH, because it affects mineral density and resorption in bones.

## 4. Discussion

Despite the lack of universal formal agreement about DDH screening, early diagnosis and management of DDH were proved to prevent long-term disabilities and provide the best clinical outcomes [9,33].

### 4.1. Description of Screening System

Neonatal hip examination (Ortolani and Barlow maneuvers) is the initial and most important part of DDH screening program [9,33,34]. Ultrasonography is considered the most reliable for screening in children less than 3 years, however, pelvic imaging is the standard method after the age of 3 years [34]. Yet, such tools might fail to detect subtle DDH [11,13].

### 4.2. Previous Reports of Screening Results

Sahin et al. [35] conducted a study to screen DDH in 5798 patients for 7 years and they concluded that serial physical examination has a very good positive predictive value in high-risk families. Gardner et al. [36] found that the use of hip U/S decreases the rates of complications and improves the quality of life and psychological health in the children. 

Two randomized controlled trials from Norway demonstrated that ultrasonographic screening helped to decrease the prevalence of late DDH presentation from 2.6–3 cases/1000 live birth to 0.7–1 cases/1000 live birth [37,38].

### 4.3. Previous Reports of the Risk Factors of Development Dysplasia of the Hip

The cause of DDH is multifactorial [3,10]. The genetic implication of DDH is now well-understood. Historically, Carter et al. [39] suggested the presence of two genetic theories for DDH; first is the polygenic inheritance to acetabular dysplasia; the second related to the autosomal dominance with incomplete penetrance. Moreover, DDH is characterized by a variable expression in the phenotype depending on the degree of femoral head uncoverage [1]. 

Kremli et al. [3] illustrated the importance of positioning and ethnic differences in the prevalence of DDH by reporting the very low prevalence of DDH in communities that carry children around the waist of mothers with the hips abducted and flexed such as India, China, and Africa. On the other hand, some studies reported that breech presentation is the single most important risk factor to develop DDH in 2–27% of children [40].

### 4.4. Descriptions of Development Dysplasia of the Hip Complications

Development dysplasia of the hip if left untreated, will lead to premature OA and long-term handicaps [1,5]. Avascular necrosis of the hip affects 60% of DDH patients due to surgical and nonsurgical intervention [41]. Williams et al. [42] reported that that risk of femoral head avascular necrosis is less than 1% with early detection and treatment with Pavlik harness. 

Studies of large families and genome-wide association studies helped to localize the genes and SNPs associated with DDH and OA. Functional SNP in *FGF-2* and *GDF5* were reported to be associated with OA secondary to DDH [16,23].

## 5. Conclusions

In 2016 Shaw et al. [43] published the new guidelines for evaluation and referral for DDH in infants, but authors still have trouble reaching consensus on diagnostic imaging in adults. The importance of more thorough research of genes associated with DDH and their variations is crucial to understanding this disorder and can lead to entirely different approach in screening of DDH. 

Genetic examination, early screening and treatment can lead to much higher effectivity in treatment and decreasing the risk of developing early OA or movement impairment, which means that, early prevention of DDH and its complications emphasizes the awareness to establish a universal screening program for newborns, that would be used internationally. Such a system, along with genetic counseling, will add value in the management of DDH as early as possible and, thus, lead to better and earlier individualized treatment. We consider screening newborns to be the most effective way to maximize the benefit for these patients. Family members of patients can benefit from such a screening program due to familiar occurrence of DDH, although such a widespread screening program will be difficult to implement at first. Problems might arise in the availability of specialized personnel and it can also prove to be an economical strain. 

## Figures and Tables

**Figure 1 medsci-07-00059-f001:**
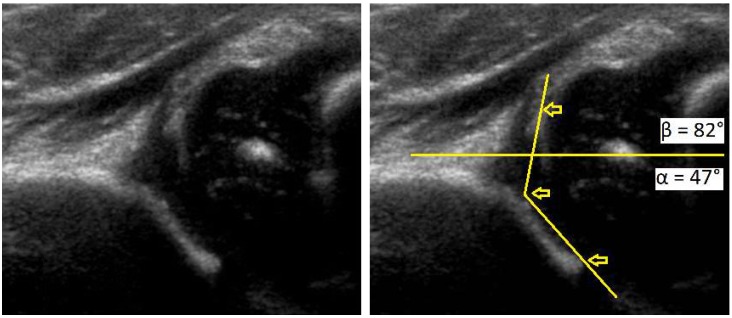
Development dysplasia of the hip (DDH)-ultrasound. In Graf’s classification type D (Decentring hip): Bony roof is severely deficient, bony rim is rounded to flattened, cartilage roof is displaced, α angle is 43–49 ° and β angle >77 °.

**Figure 2 medsci-07-00059-f002:**
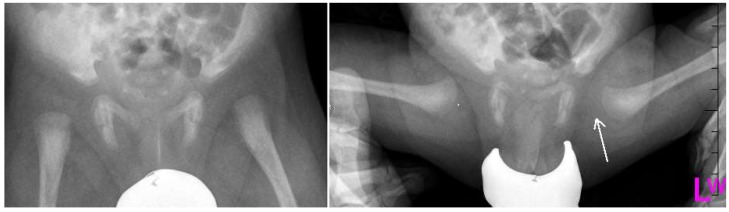
Radiographs: Antero-posterior (AP) and Lauensteun view of three weeks old child’s left hip, slightly out of the socket (eccentric hip).

**Figure 3 medsci-07-00059-f003:**
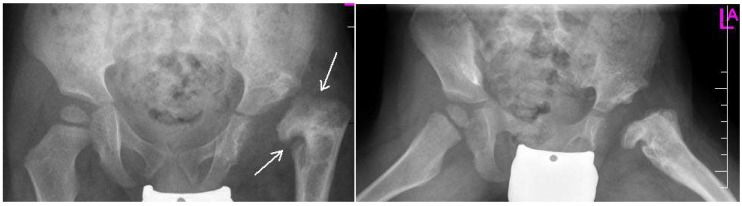
Radiographs: AP and Lauensteun with postrezidual deformity in the same patient after 2 years.

**Table 1 medsci-07-00059-t001:** List of the gene variants associated with DDH.

Genomic Design	Polymorphism/Mutations	Chromosomal Location	Genes	Location	References
GWAS	rs6060373	20q11.22	*UQCC*	China	[16]
CGAA	rs143383	20q11.22	*GDF-5*	China	[23]
CGAA	Aspartic acid (D) repeats	9q22.31	*ASPN*	China	[2]
CGAA	rs711819	2q31.1	*HOXD9*	China	[22]
CGAA	rs3744448	17q23.2	*TBX4*	China	[17]
CGAA	rs276252	20q11.22	*PAPPA2*	China	
CGAA	rs113647555	17q21.33	*COL1A1*	China	[21]
CGAA	rs1800796	7p15.3	*IL-6*	Croatia	[20]
CGAA	rs1800470	19q13.2	*TGF B1*	Croatia	[20]
CGAA	rs3732378	3q22.2	*CX3CR1*	America	[24]
GWLA	Unknown	17q21.31–17q22	*17q21.31–17q22*	China, America	[25]
GWLA	Unknown	13q22	*13q22*	Japan	[26]

GWAS: Geniomic Wide Association Studies; CGAA: Candidate Gene Association Analyses; GWLA: Genome Wide Linkage Analyses.

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
