# Peer review of "Developmental Dysplasia of Hip: Perspectives in Genetic Screening"

_medsci, 2019, doi:10.3390/medsci7040059_

Round 1
Reviewer 1 Report
add reference line 52
169- "Due to trouble with reaching a radiological consensus of DDH diagnostic in adulthood (achieved skeletal maturity) diagnosing this disorder can be problematic" - there is a clear consensus of radiological DDH diagnostic.- authors should explain or change this statement.
169- Who should be involved in a genetic screening? It seems that the authors suggest adults. If yes why?
177 indicate a weak point of the screening program - e.g. cost, availability.
Author Response
Dear editor,
we would like to thank reviewerfor comments and suggestion. All of them were used to revise manuscript.
- add reference line 52 - Suggested reference was added;
- 169- "Due to trouble with reaching a radiological consensus of DDH diagnostic in adulthood (achieved skeletal maturity) diagnosing this disorder can be problematic" - there is a clear consensus of radiological DDH diagnostic.- statement was modified;
- 177 indicate a weak point of the screening program - e.g. cost, availability - .we indicated weakpoints in the text
Reviewer 2 Report
Major comments
The abstract needs to be rewritten in its entirety.
1- The manuscript, in my opinion, need major revision. Authors failed to differentiate different study designs.
For instance, under the heading of genome-wide association studies, they mentioned genetic linkage analysis.
2- Page 2, line 92, "chromosomal regions segregating with phenotype in large pedigrees" have been detected using linkage study design and not through GWAS.
3- Page 2, line 93, the sentence "DDH inheritance is very ......" is irrelevant and not in the context.
4- Page 2, line 95, the study design used is again linkage analysis and not association study.
5- Page 3, line 107, "global CNVs detection" methods of detection have not been discussed. Please describe the study design for CVNs detections.
6-Page 3, line 127, "identified those variants ....." which variants? please discuss.
Minor comments
7-page 2, line 72, replace "aggregating" with "segregating".
8-Page 3, line 130, Table 1, what is meant by CGAA?.
9-Page 4, line 165, replace "genome association studies" with "genome-wide association studies".
8-Page 4, line 174, replace "genetical" with "genetic".
Author Response
Dear editor,
we would like to thank reviewerfor comments and suggestion. All of them were used to revise manuscript. All changes are marked by red color.
- the abstract was rewritten
- in the text, we corrected information about studies
- irrelevant text was deleted
- information about variants were added
- all minor comments were used
Round 2
Reviewer 1 Report
none